# HBV Vaccines: Advances and Development

**DOI:** 10.3390/vaccines11121862

**Published:** 2023-12-18

**Authors:** Faisal Mahmood, Ruixian Xu, Maher Un Nisa Awan, Yuzhu Song, Qinqin Han, Xueshan Xia, Jia Wei, Jun Xu, Juan Peng, Jinyang Zhang

**Affiliations:** 1Molecular Medicine Research Centre of Yunnan Province, Faculty of Life Science and Technology, Kunming University of Science and Technology, Kunming 650500, China; faisalmahmood@kust.edu.cn (F.M.); 20231118002@stu.kust.edu.cn (R.X.); yuzhusong@kust.edu.cn (Y.S.); hanqq@kust.edu.cn (Q.H.); oliverxia2000@aliyun.com (X.X.); 2Central Laboratory, Liver Disease Research Center and Department of Infectious Disease, The Affiliated Hospital of Yunnan University, Kunming 650021, China; weijia19631225@163.com; 3Department of Neurology, The Affiliated Hospital of Yunnan University, No. 176 Qingnian Road, Kunming 650021, China; maherawanmzd@gmail.com (M.U.N.A.); neurojun@126.com (J.X.); 4The Obstetrical Department, The First People’s Hospital of Yunnan Province, Kunming 650032, China; janepengjuan@163.com

**Keywords:** HBV, DNA vaccines, therapeutic vaccine, mRNA vaccine, yeast-derived vaccines

## Abstract

Hepatitis B virus (HBV) infection is a global public health problem that is closely related to liver cirrhosis and hepatocellular carcinoma (HCC). The prevalence of acute and chronic HBV infection, liver cirrhosis, and HCC has significantly decreased as a result of the introduction of universal HBV vaccination programs. The first hepatitis B vaccine approved was developed by purifying the hepatitis B surface antigen (HBsAg) from the plasma of asymptomatic HBsAg carriers. Subsequently, recombinant DNA technology led to the development of the recombinant hepatitis B vaccine. Although there are already several licensed vaccines available for HBV infection, continuous research is essential to develop even more effective vaccines. Prophylactic hepatitis B vaccination has been important in the prevention of hepatitis B because it has effectively produced protective immunity against hepatitis B viral infection. Prophylactic vaccines only need to provoke neutralizing antibodies directed against the HBV envelop proteins, whereas therapeutic vaccines are most likely needed to induce a comprehensive T cell response and thus, should include other HBV antigens, such as HBV core and polymerase. The existing vaccines have proven to be highly effective in preventing HBV infection, but ongoing research aims to improve their efficacy, duration of protection, and accessibility. The routine administration of the HBV vaccine is safe and well-tolerated worldwide. The purpose of this type of immunization is to trigger an immunological response in the host, which will halt HBV replication. The clinical efficacy and safety of the HBV vaccine are affected by a number of immunological and clinical factors. However, this success is now in jeopardy due to the breakthrough infections caused by HBV variants with mutations in the S gene, high viral loads, and virus-induced immunosuppression. In this review, we describe various types of available HBV vaccines, along with the recent progress in the ongoing battle to develop new vaccines against HBV.

## 1. Introduction

Chronic hepatitis B viral (HBV) infection affects about 296 million people worldwide and is the leading etiology of cirrhosis and liver cancer [1]. HBV can cause hepatocellular carcinoma even without developing cirrhosis as it is a DNA virus and is integrated into the human genome [2,3]. Given that HBV is highly endemic in East Asia, efforts should concentrate on preventing HBV transmission [1]; global prevalence of HBV is shown in Figure 1. By ensuring that the potential host has sufficient surface antibodies against HBV (anti-HBs), immunization against HBV can effectively prevent infection. The probability of HBV infection and the prevalence of HBV carriers are usually reduced by higher levels of anti-HBs antibodies in the circulation and a strong immune response of memory B cells triggered by the HBV vaccine. Thus, to protect the entire population, it is necessary to administer a universal HBV vaccine without fail, while also monitoring anti-HBs levels after vaccination. However, people who are regularly exposed to HBV, such as healthcare workers, are at increased risk [4]. As a result, follow-up plans may need to be developed for patients who are at a high risk of transmitting HBV. According to some findings, HBV transmission can occur when anti-HBs titer levels are inadequate [5]. Several approaches have been attempted to develop a therapeutic vaccine to prevent chronic hepatitis B. Most of the time, they were established on advanced understanding about the immunopathogenesis of HBV infection and novel techniques for enhancing the potency of virus-specific immunity [6]. Several clinical trials have been conducted in patients with chronic hepatitis B to investigate the effect of certain ideas on the restoration of protective immunity after the establishment of persistent HBV infection. The goal of preventive vaccinations is to create immunity against HBV surface antigens to produce antibodies that destroy HBs antagonists and prevent infections. The acceptance of HBs-explicit CD8 immune system, that target and wipe out HBsAg infected hepatocytes is less significant.. On the other hand, in therapeutic vaccination, potent CD4 and CD8+ T cell responses are primarily targeted, while other viral antigens, such as the HBcore antigen and the viral polymerase, are also targeted to broaden the scope of the virus-specific effector T cell response. Reduction in viral replication was a common component of all attempts for developing therapeutic vaccination against chronic hepatitis B [7].

## 2. Hepatitis B Virus

HBV is a vaccine preventable disease that increases the economic burden on families, communities, and countries [9]. Given the high incidence of viral hepatitis B, prevention is the only effective method of treatment [10]. According to a global systematic assessment, approximately 248 million people worldwide had a chronic HBV infection in 2010. China has been classified as a higher intermediate-endemic country, with 74 million HBsAg-positive individuals, or 5.49 (5.47–5.50) percent of the total Chinese population [11]. There are significant differences in the prevalence of HBV not just across regions and within countries, but also by state or province, income, race or ethnicity, and other social and cultural variables [1]. The burden of HBV infection is highest in the World Health Organization (WHO) Western Pacific Region and the WHO African Region, where 116 million and 81 million people, respectively, are chronically infected. A total of 60 million people are infected in the WHO Eastern Mediterranean Region, 18 million in the WHO South-East Asia Region, 14 million in the WHO European Region, and 5 million in the WHO Region of the Americas [12]. 

HBV has a unique biological process characterized by an error-prone reverse transcriptase activity and an exceptionally high daily production of viral particles. In addition, it can easily carry out mutations that lead to the generation of different genetic variants [13]. The genome of the HBV consists of a relaxed-circular DNA (rcDNA) that is approximately 3.2 kilobases (kb) in length. This DNA molecule has a complete minus (−) strand and an incomplete plus (+) strand. The basic structure of the HBV is shown in Figure 2. Within the HBV genome, there are four open reading frames (ORFs) that overlap with each other. These ORFs are indicated as C, P, S, and X. Each of these ORFs is responsible for producing specific viral proteins. The encoded proteins resulting from these ORFs include HBc and its related proteins, such as the E antigen (HBe) and the 22-kilodalton precore protein (p22cr). Furthermore, there are the protein Pol, three types of surface antigens known as L-HBs, M-HBs, and S-HBs, and the HBV X protein (HBx). These viral proteins play important roles in the lifecycle of HBV. When a cell becomes infected with HBV, the rcDNA is transformed into covalently closed circular DNA (cccDNA). This cccDNA then generates HBV RNAs of various lengths. The most abundant lengths of these RNAs are approximately 3.5, 2.4, 2.1, and 0.7 kb. These RNAs are transcribed from different promoters within the HBV genome. The 3.5 kb RNA leads to the production of proteins from the C and P ORFs. The 2.4 kb RNA is responsible for the translation of L-HBs. The 2.1 kb RNA results in the synthesis of two other surface antigens, namely M-HBs and S-HBs. Finally, the 0.7 kb RNA produces the HBx protein. Using these viral proteins in combination with factors derived from the host cells, HBV is capable of proliferating within the host cells [14]. The hepatitis B virus-related hepatocellular carcinoma tissues expressed 17 proteins in a distinct manner, with 10 being up-regulated and 7 being down-regulated. The HBV vaccine antigen has a single immunological specificity and four antigenic epitope subtypes: two sets of mutually exclusive “d” or “y” and “w” or “r” epitopes, and group-specific “a” epitope. As a result, four serotypes with varying degrees of regional frequency were identified: adw, adr, ayw, and ayr [15]. 

HBV can only infect hepatocytes with the help of the bile acid transporter sodium taurocholate co-transporting polypeptide (NTCP) [16]. The most common ways to get infected are through sexual contact, exposure to blood contaminated with viruses (from unscreened blood transfusions or reusing contaminated needles), or vertical transmission from mother to child at birth [17]. Nucleos(t)ide analogues, the foundation of chronic HBV therapy, prevent viral replication and reduce the risk of liver cirrhosis and hepatocellular carcinoma. However, if treatment is stopped, the viral load usually increases due to the uncontrolled transcription of HBV cccDNA, which exists as a minichromosome in the nucleus of infected hepatocytes [18]. The ultimate goal of HBV therapy is “functional cure,” which is defined as a persistent loss of HBsAg. In this context, despite low levels of HBV cccDNA, a functional adaptive immune response ensures that viral replication is suppressed without the need for therapy, similar to what happens after acute HBV infection has cleared up [19]. For HBV clearance in acute infection, a potent HBV-specific CD8+ T cell response is necessary, but in chronic HBV, the T cell response is defective and only partially restored by nucleos(t)ide analogues [20]. 

## 3. HBV-Induced Immune Dysregulation

The immune system needs a robust and wide-ranging T-cell immunological response in order to eradicate HBV, and chronic HBV infection usually leads to a dysregulated immune response. The host’s inability to eradicate the enduring infection is thought to be mostly due to the depletion of HBV-specific T lymphocytes. Using therapeutic vaccination to increase the patient’s own cellular immune response against viruses is seen to be a viable approach. Several attempts have been undertaken over the past 20 years to develop a therapeutic vaccination against chronic hepatitis B (CHB) [21]. Currently available prophylactic HBV vaccinations work by triggering humoral immune responses against HBsAg, which neutralizes infectious HBV particles before they can enter hepatocytes. Although prophylactic vaccination against HBV requires antibodies, the failure of approved prophylactic vaccines in therapeutic vaccination attempts suggests that this immune mechanism alone will probably not be enough to eradicate HBV-infected cells in that scenario [22].

Numerous pattern recognition receptors are expressed on the surface of human innate immune system cells. When they identify patterns associated with infections or trauma, they set off an internal signaling cascade that eventually releases pro-inflammatory cytokines and antiviral interferons (IFNs). Previous studies on chimpanzees and humans revealed that type I IFN production was minimal during the early phases of viremia, indicating that HBV functions as a “stealth virus” [23]. Furthermore, there is an attraction of myeloid-derived suppressive cells to CHB, which further enhances the chain reaction of immunosuppression. These cells secrete IL-10 and arginase, effectively hindering the expression of T cell IFN-α [24]. Consequently, HBV induces the infiltration of IFN-α deficient natural killer cells and T regulatory cell populations in the liver, resulting in a tolerogenic microenvironment [25]. The development and differentiation of T and B cells are significantly impacted by this dysregulated environment, impairing adaptive immune responses. Diminished cytotoxic capacity, impaired proliferative capacity, and heightened levels of inhibitory molecules (such as CTLA-4, PD-1, T cell immunoglobulin, and mucin-domain containin-3) are also indicative of T cells becoming depleted after prolonged exposure to high concentrations of viral antigens. Dysregulation of immune response after HBV infection is shown in Figure 3. In order to effectively address chronic infections in patients, it is necessary to devise strategies that can rejuvenate or enhance enduring HBV-specific cellular and humoral responses. Recently, there have been efforts and ongoing research in employing immunotherapies to manage CHB infection in a better way. The compromised HBV-specific T cell response can potentially be restored through immunotherapy techniques involving vaccinations and checkpoint inhibitors, given their ability to bolster T cell activities in laboratory settings [26]. However, it is yet unclear whether these methods will be effective in inducing long-term HBV control and restoring antiviral T cell immunity. 

## 4. Hepatitis B Vaccines

In the early 1980s, the United States and France worked together to produce the first hepatitis B vaccinations. These vaccines, also referred to as plasma-derived vaccines, were prepared by extracting minuscule particles of HBsAg measuring 22 nm from the blood of carriers. To ensure their safety and purity, these particles were subsequently treated with urea, pepsin, formaldehyde, and heat, which inactivated harmful elements and refined them. These pioneering immunizations have successfully reached millions of individuals, garnering a remarkable record of reliability and efficacy [27]. Despite achieving notable success, concerns have been growing in Europe and North America regarding the potential contamination of the plasma-derived vaccines with other bloodborne viruses like HIV. It is important to note that HBV and HIV share certain common modes of transmission. Since recombinant DNA (rec-DNA) vaccines were mostly developed by the mid-1980s, the use of these vaccinations has been restricted due to misleading safety concerns. It is important to emphasize that there are no known incidences of HIV, or any other illnesses associated with hepatitis B vaccination.

Using this modern approach, second-generation vaccines have been developed by producing the HBsAg protein within genetically modified mammalian cells or yeast cells (*Saccharomyces cerevisiae*) containing the HBV surface gene (S gene) [27,28]. The production of yeast-specific antigens involves two steps: the S gene is isolated from HBV and introduced into yeast cells, and the yeast cells were grown by fermentation process to produce HBsAgs. The HBsAg polypeptides obtained after extraction and purification, self-assemble into small, non-pathogenic particles similar to the 22 nm particles found in the blood of HBsAg carriers. Afterwards, the HBsAg is adsorbed on aluminum hydroxide (Al (OH)_3_), which acts as an adjuvant to increase the immunogenicity of the vaccine. The manufacture of vaccines may be expanded worldwide and continue indefinitely with the application of this particularly mature, new genetic engineering technique. Both host factors (such as age, multiple chronic conditions, prior HBV exposure, and time since vaccination) and vaccine-related factors (such as the kind, dose, and timing of the vaccine delivered) have an impact on the vaccine’s efficacy [29].

In the 1990s, researchers developed third-generation HBV vaccines using HBV-infected mammalian cells. These cells included Chinese hamster ovary (CHO) cells and mouse cell lines expressing and releasing small S and medium pre-S2 glycoproteins. In some cases, all three viral envelope proteins (S, pre-S1, pre-S2) were included. When compared to vaccines derived from yeast that only contain the S protein, these new vaccines have demonstrated promising results due to their capacity to elicit a stronger and quicker antibody response. Thus, pre-S/S protein-containing vaccines may be more effective in immunizing individuals who have previously shown poor or no response to standard vaccines [30]. Recent studies have found that a hepatitis B vaccine with three antigens (known as TAV) consisting of S, pre-S1, and pre-S2 antigens was more effective compared to a vaccine with only one antigen (known as MAV) containing S antigen. This superior performance of the TAV vaccine was observed in terms of inducing a strong immune response in both healthy adults aged 18 years and above, as well as individuals aged 45 years and older [31]. Furthermore, the rapid and high rates of seroprotection that were achieved after TAV delivery suggest that this vaccine can offer seroprotection to a greater number of individuals than MAVs. These findings may have clinical implications, particularly for the large proportion of adult non-responders to standard hepatitis B vaccinations, those who are more likely to contract the disease, and those who are more likely to experience severe complications from HBV infection [32]. Different vaccine development methods are given in Table 1.

Since then, a number of combination vaccinations have been developed to make it easier to incorporate the HBV vaccine into immunization programs that were first created for newborns and early children. In October 2000, the hexagonal vaccine, also known as the six-in-one vaccine, was authorized by the European Union for the primary immunization of infants within their first year of life. This involved administration of three doses 1–2 months apart, starting from the age of 2 months, with a booster dose given 6 to 12 months later. These vaccines offer protection against polio, diphtheria, tetanus, pertussis, hepatitis B, and infections caused by *H. influenza* type B. Extensive research has shown that each of these combinations retains antigenic components that are highly immunogenic, safe, and well-tolerated [37]. This makes immunization schedules more manageable by allowing for the delivery of broadly efficient vaccines with fewer injections [38]. Combination vaccinations that contain the antigens for both hepatitis A and hepatitis B have been successfully developed in order to provide dual protection against both illnesses. Those who are more vulnerable to contracting hepatitis A and B, such as military personnel and tourists from areas where these illnesses are highly prevalent, will benefit greatly from the use of these vaccines. It has been demonstrated that these vaccinations’ efficacy and safety are comparable to those of administering hepatitis A and B monovalent vaccines on their own [39,40].

New adjuvants have been developed to enhance the immune response against HBsAg in certain groups of individuals who do not respond or have a poor response to traditional hepatitis B vaccines [41]. Many countries have produced and used the HBsAg-enhanced immunogenicity vaccine adjuvanted with aluminum phosphate and 3-O-deacyl-4′-monophosphoryl lipid A (MPL). It includes immunocompromised individuals, such as those with renal failure, and individuals who have undergone liver transplantation [42]. A recombinant monovalent HBV vaccine adjuvanted with CpG (cytosine phos-phoguanosine), which can activate innate immunity through TLR9, was recently licensed by the FDA and European Medicines Agency (EMA) [43]. This new vaccine requires two doses within 1 month and is currently only approved for adults. This vaccine offers another alternative to HB immunization in hypo/non-responders due to a shorter schedule, early seroprotection, better infectivity and high seroprotection. Various certified adjuvant vaccines are shown in Table 2.

### 4.1. Needle-Free Hepatitis B Vaccine

The term “mucosal vaccines” has been used to describe oral, intranasal, pulmonary, rectal, and cervical injections. Undoubtedly, the mucous membrane, with a total surface area of about 400 m^2^, is the main route through which pathogens enter the body [47]. As mucosal surfaces are the main entrance points for bacteria and viruses, mucosal vaccination would be the first line of defense because it encourages the secretion of IgA, which keeps infectious pathogens from sticking to the mucosa. Additionally, despite the fact that hepatitis B vaccines currently on the market are regarded as safe and efficacious, maintaining efficacy requires three doses, which decreases patient compliance. Furthermore, well-planned vaccine storage facilities and qualified medical personnel are needed, which could be difficult, especially in developing countries. Considering these facts, the effectiveness of vaccination against HBV could be significantly impacted by needle-free vaccination [48]. Some people have high hopes for mucosal vaccinations from a variety of angles. One of the most important benefits is that the local mucosal immune response is critical for preventing early infection, particularly for diseases with mucosal surfaces as their site of origin, such as sexually transmitted HBV. Traditionally administered vaccinations are often ineffective at generating mucosal immunity and instead mainly stimulate systemic immune responses. On the other hand, mucosal vaccinations that are taken orally, nasally, or sublingually typically elicit systemic as well as mucosal immune responses. Administration of vaccines through mucosal surfaces may stimulate the synthesis of pathogen-specific mucosal immune responses [49]. Additionally, mucosal immunization is frequently viewed as a desirable alternative to parenteral immunization since it can concurrently generate mucosal and systemic immunity and promote both humoral and cell-mediated immune responses [50]. 

Saraf et al. developed and improved lipospheres for delivering HBsAg intranasally [51]. A recent study have investigated whether surface-modified liposomes might be used for nasal vaccination [52]. The researchers created liposomes coated with glycol chitosan and loaded them with DNA complexes containing carrying pRc/CMV-HBs. Since chitosan has mucoadhesive qualities [53], it is preferable to include this polymer in the construction to produce an intranasal formulation with long-lasting local retention. Bilosomes were investigated to develop an oral hepatitis B vaccine, due to their demonstrated potential for application in oral vaccination carrier systems [54,55]. They showed stability at different bile salt concentrations, in simulated intestinal fluid (SIF), and in simulated gastric fluid (SGF). Niosomes, which are unilamellar or multilamellar vesicles generated from synthetic, non-ionic surfactants of the alkyl or dialkyl polyglycerol ether famifly, offer an alternative to liposomes as drug carriers due to their superior chemical stability [56]. Although the ability of these non-ionic surfactant vesicles to act as adjuvants has long been established, only recently they have been used to develop a new HBV oral vaccine delivery strategy [57]. Makidon et al. tested a nanoemulsion-based hepatitis B vaccine on nasal mucosa. They assessed the immunological response produced by the vaccine formulation, which included NE and 20 µg of recombinant HBsAg. The positive control was a vaccine administered intramuscularly and adjuvanted with aluminum hydroxide. In contrast to HBsAg-NE, which induced the production of IgG2b (and some IgG2a) antibodies mainly associated with T-helper cell 1 (Th1) immune reactions, the aluminum-formulated vaccine induced a greater production of IgG1-class antibodies, which correlate with Th2 immune responses. However, both vaccines yielded comparable levels of Anti-HBsAg IgG production [58], and nano-adjuvants can produce a more balanced Th1/Th2 reaction to compensate the weak Th1 reaction of aluminum hydroxide adjuvants [59].

### 4.2. Therapeutic Vaccine

The term “therapeutic vaccination” refers to immunizing people against a non-infectious version of viral antigen in an effort to accelerate or enhance preexisting HBV-specific immunity reactions, leading to the long-term management of HBV infection. Early research on therapeutic vaccination focused on prophylactic recombinant vaccines, such as those containing HBsAg-expressing HBV envelope proteins. Considering that the two main objectives of any anti-HBV therapy are the removal of HBsAg and the generation of protective anti-HBs antibodies, the initial ruling seemed logical. Clinical pilot studies show that receiving the recommended vaccine significantly increased HBeAg seroconversion rates. Researchers have used different approaches to create therapeutic vaccines for HBV using one or more hepatitis B virus proteins on different platforms in combination with targeted therapy [60]. Schematic presentation showing basis behind therapeutic vaccine is given in Figure 4.

Therapeutic vaccines for chronic HBV infection have shown great difficulty in restoring robust HBV-specific immunity. The goal of therapeutic vaccination is to stimulate new or enhance existing HBV-specific T cell responses by directly delivering non-infectious HBV antigen to chronically infected individuals [61]. The ability of the host to generate effective B and T cell responses specifically adapted to HBV greatly influences the outcome of HBV infection. After infection, people can develop an acute, self-limiting HBV infection characterized by strong antibodies and robust T cell responses. Alternatively, they may face a chronic infection characterized by rare, often impaired T cell responses and a lack of HBV-specific neutralizing antibodies [62]. Therapeutic immunization represents a promising treatment strategy to effectively combat the virus and ultimately cure the disease by enhancing the body’s immune response specifically targeted against the HBV [63]. TherVacB, a novel therapeutic hepatitis B vaccine, aims to overcome HBV-specific immune tolerance and restore antiviral T-cell and B-cell responses to treat HBV. It is a vaccination method using a heterologous protein preparation combined with modified vaccinia Ankara (MVA) retreatment. This innovative strategy aims to trigger the activation of helper and effector T cells and HBV-specific B cells based on the latest research findings [64,65]. TherVacB has the ability to disrupt the natural immune tolerance to HBV in experimental models of HBV transgenic mice and AAV-HBV-infected mice. By initiating the activation process through the presentation of particulate HBsAg and HBV core antigen (HBcAg), TherVacB effectively activates HBV-specific B cells, CD4+ helper T cells, and CD8+ effector T cells. The subsequent MVA-vector boost is designed to further enhance the functionality of CD8+ effector T-cells that specifically target HBV. As a result, these multifunctional and multispecific T cell responses play a crucial role in regulating HBV infection by eliminating virus-infected liver cells. In addition to T cell responses, TherVacB also induces the production of antibodies that block the spread of the virus [66]. In preclinical trials, the therapeutic HBV vaccination VRON-0200 containing an intrinsic checkpoint inhibitor, produces widespread CD8+ T cell responses and long-lasting antiviral effects [67]. VRON-0200 vaccination elicits potent and broad CD8+ T cell responses to HBV core and polymerase, and it induces CD8+ T cells traffic to the liver. VRON-0200 vaccine-induced CD8+ T, but not CD4+ T cell responses correlate with antiviral activity [67,68]. 

Restoration of effective antigen specific CD8+ T cell responses is a primary goal of many immunotherapies being developed for chronic viral infections and malignancies. To date, therapeutic vaccines have had limited success, but there remain important strategies to enhance the endogenous immune response to specific antigens. However, proliferation of existing T cells can be limited by their profound exhaustion, while newly generated responses can suffer from corresponding tolerogenic effects that can limit their expansion and survival. The more immunogenic therapeutic vaccines already in development will likely need to be used in combination with specific immunotherapies to enhance the T-cell response [69]. A functional cure has been shown to be induced by therapeutic vaccination in animal models of chronic HBV, with the goal of reestablishing the HBV-specific immune response [62]. The challenges to immunological control posed by chronic infection may actually be overcome by optimizing T cell responses through the selection of maximally immunogenic vaccine vectors, vaccination routes, and immunoprotected epitopes in conjunction with agents that lessen HBV T cell dysfunction, such as immune check-point inhibitors. Clinical trials of ChiCTR2300071992 vaccine are undergoing to evaluate its safety and efficacy in patients with chronic hepatitis B [70]. ChiCTR2200065085 is also in clinical trials for the evaluation of the tolerability, safety and preliminary efficacy of vaccine in patients with chronic hepatitis B [71]. ISA104 is also under investigation towards the effectiveness, safety, and tolerability of various dosages in comparison to placebo in individuals with chronic HBV. The research’s details are available at clinicaltrials.gov with the study number NCT05841095 [72].

### 4.3. Adenovirus Vaccines

For over four decades, recombinant viral vectors have been used as a vehicle to transport antigens associated with particular illnesses [73,74]. One advantage of viral vectors over traditional subunit vaccines is that they not only trigger potent antibody responses but also cellular responses that are crucial for the eradication of pathogen-infected cells. Additionally, viral vectors frequently result in considerable immunogenicity and long-lasting immune responses even without the use of an adjuvant [75].

Throughout history, the main focus of vaccine development has been the use of attenuated forms of specific pathogens or their protein components. Although these methods have proven to be effective protection against many dangerous diseases. Advancements in vaccine technology have led to introduction of nucleic acids and viral vectors as very promising alternative [76]. Researchers have been observing the use of adenoviruses (AdVs) as vehicles for administering vaccines for over three decades. The ability of AdVs to elicit both innate and adaptive immune responses in recipients is not advantageous for most therapeutic treatments; nevertheless, this fundamental characteristic is highly advantageous in the production of vaccines [77,78].

It is important to note that there is considerable variation in the use of AdVs for HBV immunization. However, progress has been made in the use of AdVs in immunotherapies and HBV-specific therapeutic vaccination strategies. Activation of the immune system is necessary to establish an effective defense against viral infections. One notable example is TG1050, an innovative anti-HBV immunotherapeutic derived from Adenovirus 5 (Ad5). It contains a fusion protein that comprises modified HBV core, polymerase, and specific envelope protein domains. Administering a single dose of TG1050 resulted in an increased generation of HBV-specific T lymphocytes in the liver and spleen, which triggered the destruction of cells and reduced the amount of viral replication markers present in circulation [79]. A recent study by Chinnakanna et al. presents groundbreaking research on the development of therapeutic viral vectors, namely ChAdOx1 and MVA for the treatment of HBV. ChAdOx1 contains a genetic sequence that encodes an HBV immunogen consisting of three complete HBV antigens: pre-core/core, polymerase, and surface. They administered ChAdOx1 to mice with a fully functioning immune system, but without HBV infection, resulting in enhanced T- cell responses. Subsequently, the mice received a heterologous MVA-boost vaccination. The combined action of polyfunctional CD8+ and CD4+ T cells produced a variety of cytokines, including IFNs, TNF-α, and IL-2. Overall, this study highlights the potential of these therapeutic vectors in eliciting a robust immune response against HBV. The success of immunotherapeutic approaches using anti-HBV AdV underscores the remarkable ability of AdVs to efficiently deliver HBV antigens into cells [80].

Adenoviruses are in high demand in the development of HBV-targeted gene therapies due to their strong affinity for the liver. The replication intermediate of HBV virus has great potential to create an effective therapy to eradicate HBV infection, because survival of the virus and number is highly dependent on functional cccDNA. Promising results have been obtained using TALEN- or CRISPR/Cas9-based therapies to specifically target cccDNA [81,82].

### 4.4. mRNA Vaccines

mRNA vaccines offer a compelling alternative to traditional vaccine technologies because they allow rapid and scalable production without the risk of infection or integration. These vaccines successfully induce the desired immune responses and have a stronger ability to stimulate the innate immune system compared to plasmid vaccines [83]. Lamb et al. [84] have developed and implemented a technique to synthesize mRNA containing all the necessary HBsAg components to create an mRNA vaccine for the prevention or treatment of HBV. This approach involves the design of mRNA-lipoplex nanoparticles capable of preventing degradation by exonucleases, facilitating cellular uptake by endocytosis, and allowing release of mRNA from endosomes. Similar to DNA vaccines, mRNA-derived antigen products are processed and transported to the target site, stimulating strong humoral and cell-mediated immune responses against the pathogen [84,85]. It is worth mentioning that a very reliable method was employed to create a vaccine formulation based on mRNA. This formulation successfully resulted in the production of detectable L-HBs and S-HBs in cultured hepatoma cells. Although the secretion was moderate and the expression of L-HBs was lower than expected, it is possible that this could have reduced its ability to stimulate the immune system [84].

mRNA-based vaccination is a technique for generating strong humoral and cell-mediated immune responses that are unique to the antigen. mRNA immunotherapy was performed by removing dendritic cells (DCs) from a patient, modifying them ex vivo with the mRNA construct, and reintroducing them to the patients [85]. Ex vivo DCs engineering is an expensive and complex approach of personalized therapy that does not withstand clinical viability and safety. The difficulties listed above considerably limit its use among the general public [86]. However, mRNA-based vaccinations have not been extensively studied in the context of HBV prevention or control methods. 

### 4.5. Yeast-Derived Vaccine

Unicellular fungi and yeast with highly efficient systems for expressing genes from other organisms include *S. cerevisiae*, *Schizosaccharomyces pombe*, *Saccharo-myces boulardii*, *Hansenula polymorpha*, *K. phaffii*, *Candida boldmu*, *Kluyveromyces lactis*, and *Yarrowia lipolytica*. The ability of Blastobotrys (*Arxula adeninivorans*) to utilize different substrates, the accessibility of its genomic sequence, and its receptiveness to genetic modification make it a crucial host for expressing foreign proteins, in addition to the aforementioned yeast species [87]. The advancement of rec-DNA technology has facilitated the expression of HBsAg in yeast and subsequently in mammalian cells, thus enabling the large-scale production of vaccines [88]. The strategy currently used for the production and implementation of yeast HBV vaccine is shown in Figure 5. 

The most popular hepatitis B rec-DNA vaccines are made from yeast by expressing the HBsAg protein in yeast cells (*S. cerevisiae*) that have been genetically modified to contain the S gene. Several other species of yeast used for production of HBV vaccines are shown in Table 3. HBsAg self-assembles into virus-like particles (VLPs) when administered as a vaccine, thereby protecting against HBV infections. The recombinant hepatitis B vaccines mark a crucial turning point in the history of immunization since they were the first vaccines constructed on VLPs. The yeast *S. cerevisiae* can produce HBsAg that can combine to form particles that resemble the 22-nm particles produced by humans [89]. The hepatitis B vaccine, consisting of HBsAg derived from genetically engineered yeast cells, successfully triggered an immune response against anti-HBs in mice, monkeys, and chimpanzees. Furthermore, the chimpanzees that were administered the vaccine showed complete immunity when exposed to intravenous doses of either identical or different strains of human HBV [90]. 

Several human clinical trials have demonstrated that the rec-DNA yeast hepatitis B vaccine is not only safe but also has properties comparable to human plasma vaccination. These characteristics include similar quantity, quality, specificity and protective efficacy of the anti-HBs response [91]. Due to the unlimited availability of HBsAg produced by recombinant DNA technology and HBsAg produced in yeast, which have all the properties of natural HBsAg in human plasma except glycosylation, reducing the hepatitis B vaccine was no longer considered a rational decision [92].

**Table 3 vaccines-11-01862-t003:** Table showing list of HBV proteins expressed in various yeast species.

Yeast Species	Antigen/Immunogen	Expression Strategy	References
*S. cerevisiae*	HBsAg	Virus like particle	[93]
	Hepatitis B surface antigen	Purified protein	[94]
	Hepatitis B surface antigen (HBsAg)	Virus like particle	[95]
	Hepatitis B virus (HBV) X, surface(S), and Core antigens (X-S-Core)	Whole recombinant yeast	[96]
	Surface protein GS-4774	Whole recombinant yeast	[97]
	HBsAg	Virus like particle	[98]
	Hepatitis B core protein	Purified protein	[99]
	Hepatitis B surface antigen	Purified protein	[100,101]
*P. pastoris*	HBsAg and HEnAg fusion protein	Virus like particle	[102]
	HBsAg	Virus like particle	[103]
	Core protein (HBc)	Virus like particle	[104]
	Recombinant hepatitis B surface antigen	Purified protein	[89]
	HBsAg, HSP70 (1–370)	Whole recombinant yeast	[105]
	HBsAg	Virus like particle	[106]
	Hepatitis B surface antigen	Virus like particle	[107]
	Recombinant hepatitis B surface antigen	Virus like particle	[108]
*H. polymorpha*	VrHB-IB	Purified protein	[109]

Yeast-derived HBV vaccination has been used worldwide. Japan presently uses two HBV vaccines that make use of S-HBsAg made from yeast representing different various HBV genotypes. Both are likely to have acceptable safety profiles and an equally effective ability to elicit neutralizing antibodies [110]. However, a number of issues have been observed that cannot be ignored. For instance, the humoral response to these vaccines is reported to be weak or absent in 10% of people who had vaccinations [111]. Variations in the amino acid composition of these variants are thought to be responsible for the virus and its ability to evade neutralization by antibodies produced by HBV vaccines [112]. These variations are commonly referred to as vaccine-escape mutations (VEMs). However, a comprehensive and accurate assessment of the effect of these amino acid variants on the neutralizing efficiency of HB vaccine-induced antibodies has not yet been achieved [113].

Newer therapeutic vaccine candidates have been developed with the aim of inducing a diverse range of immune responses against HBV by integrating different HBV antigens or epitopes. One such vaccine, GS-4774, derived from yeast, incorporates key components for HBsAg, HBcAg, and HBX. These specific elements were selected due to their highly conserved nature across various HBV genotypes [97]. Successive trials conducted on individuals suffering from persistent HBV infection have yielded disappointing outcomes. In a comprehensive study involving 178 patients with chronic HBV infection, with attenuated HBV DNA and relatively low HBsAg levels (average at 2.9 log10 IU/mL), evaluated the efficacy of nucleos(t)ide analogue treatment both by itself and in combination with GS-4774 immunization. Although there was a decrease in HBsAg levels observed across all groups, only three out of the 178 individuals experienced a significant drop of 0.5 log10 IU/mL or more. Remarkably, all three individuals belonged to the vaccination groups that received the highest doses. Upon completion of the trial, an immunological analysis detected modest T cell responses post-vaccination [114]. Subsequently, in a group of 195 individuals with chronic HBV infection exhibiting increased HBV DNA levels and a higher HBsAg titer (average 3.7 log10 IU/mL), the efficacy of GS-4774 was evaluated compared to a combination of tenofovir and nucleos(t)ide analogue tenofovir. None of the participants experienced a decrease in HBsAg levels even after 48 weeks of treatment, and there were no significant differences observed in the decline rate of HBsAg titers among the different treatment groups [115]. One possible explanation for the ineffectiveness of this vaccine in providing immunological protection in individuals with chronic HBV infection is its limited ability to induce significant T cell responses. Even among healthy participants, the majority showed only modest T cell responses, which may not be powerful enough to overcome the T cell tolerance associated with chronic HBV. This deficiency may have contributed to its limited efficacy observed in these studies [97].

Different strains of yeast responded differently to HBV. *H. polymorpha* has enabled the efficient production and purification of HBV’s gp96 protein, allowing for high yields. Additionally, mice that have been immunized can effectively elicit a specific immune response known as cytotoxic T lymphocytes (CTL) against HBV [116]. The clinical testing for the heat-killed whole recombinant budding yeast vaccine (GS-4774) targeting the HBV has successfully concluded its phase II trial [114]. 

### 4.6. DNA Vaccines

DNA vaccine technology is based on modified plasmids that contain an efficient promoter that increases the transcriptional activity of cells. These plasmids also carry a specific sequence that encodes the chosen immunogenic antigen(s). A suitable plasmid is administered after intramuscular injection either throughout the body or mainly on the surface. The cellular system of the host is used to transport the foreign DNA to the nucleus of the transfected cells, which includes antigen-presenting cells (APCs) found locally [117]. DNA vaccine platforms offer cost-effective production and storage due to their rapid construction, ease of replication, and exceptional stability at room temperature [118]. Given that they consistently discharge antigens, eliciting a persistent response that lacks infectivity and the possibility of causing anti-vector immune reactions, theoretically, they pose less risk than traditional live attenuated vaccines. Furthermore, the internal production of the encoded antigens allows for natural post-translational alterations, maintaining the proteins in their original form [119]. Indeed, DNA vaccination has exhibited encouraging outcomes in preclinical models investigating chronic hepatitis virus infection [120]. 

To investigate whether the long-term presence of antigens interferes with the ability of therapeutic vaccines to activate resting HBV-specific T cells, DNA vaccines were administered in combination with conventional therapies. The efficacy of a DNA vaccine expressing the PreS2/S protein was evaluated in a phase I/II clinical trial in virally suppressed patients receiving stable NA therapy. The vaccine caused a substantial increase in CD4+ T cell responses, with multiple specificities, improved functionality, and longer duration, when used in conjunction with NA therapy. However, the immune-boosting effect lacked the capacity to influence relapse rates upon cessation of NA therapy [62]. When the HBV enters the body, the immune system’s cellular and humoral responses are necessary for clearing the virus. The humoral immune response helps remove viral particles from the bloodstream and prevent the virus from spreading in the host. On the other hand, the cellular immune response is to target and eliminate infected cells. Current commercially available vaccines often fail to stimulate both branches of the immune system effectively. However, DNA-based immunization has the potential to enhance immunotherapeutic strategies by eliciting both cellular and humoral responses. This innovative vaccination method involves directly injecting a gene in the form of pure plasmid DNA, which triggers an immune response to a protein produced within the body [121,122]. DNA-based vaccination has shown a diverse array of immunological responses, including the production of neutralizing antibodies and the activation of cytotoxic and T- helper cell reactions, among other outcomes [29]. Recent research indicates that immunizing animals with DNA plasmid vectors significantly promotes a Th1 response [123]. As a consequence of the induction of Th1-biased immune responses, T-helper lymphocytes were found to produce IFN-γ in response to similar antigenic stimuli [124]. IL-2 and IFNs are examples of Th1-type cytokines that primarily link to cell-mediated immune reactions, playing a crucial role in protecting the body against intracellular pathogens such as viruses. There is a proposal that an unequal distribution of Th2 and Th1 responses could potentially be the underlying factor behind the persistent HBV infection, leading to a chronic condition [125].

### 4.7. Recombinant Vaccines

Early research primarily focused on the small and medium surface proteins of the HBV, namely Pre-S2 and S protein. These proteins have been studied in connection with the development of combination vaccines. The reason for this study is their ability to induce a strong immune response, especially the production of HBsAbs and the activation of CD4+ T cells, even in people not infected with HBV. Therefore, scientists developed preventative immunizations in the form of recombinant HBsAg vaccines to provide protection against HBV infection [126]. However, upon examining the impact of vaccination on individuals suffering from untreated chronic HBV infection, no apparent difference was found between the vaccinated and unvaccinated groups in their ability to achieve HBV DNA suppression, HBeAg loss, or HBsAg loss, despite the fact that vaccination did elicit detectable surface-specific T cell responses in a portion of individuals [127]. In a great effort to enhance the body’s immune response, researchers examined the combination of HBsAg with hepatitis B immunoglobulin in untreated individuals. Unfortunately, this approach yielded similarly discouraging results [128,129]. Comparison of vaccine therapy with nucleoside analogues or interferon therapy did not show additional benefits [130,131]. Phase II clinical trials are currently underway for CVI-HBV-002, an alternative recombinant vaccine designed for individuals afflicted with chronic HBV. Although the findings of these trials have not been disclosed to the public as of now, this vaccine contains the crucial components of long, medium, and small HBV surface proteins, specifically targeting the surface, pre-S2, and pre-S1 regions [62,132].

One possible explanation for the failure of these trials is the assumption that vaccination methods needed to cure established chronic HBV infection would be significantly different from those required to successfully prevent HBV infection. It is anticipated that the use of protein-based vaccines alone would have a minimal impact on T cell production and immune system restoration in the chronic environment, despite the fact that HBsAg triggers the production of HBsAbs, which can block viral entry and prevent infection in healthy individuals. Another reason for the unsuccessful results of these trials may be that many participants with elevated HBV antigen levels, were included in the initial trials of the recombinant HBsAg vaccine. It has been found that high antigen levels act as a barrier to the effective recovery of tolerized T cells through therapeutic vaccination [21]. Two small trials have assessed the effectiveness of the recombinant HBsAg vaccination in individuals who have a disease that lacks the specific antigen, suppressed HBV DNA, and low levels of antigens (between 100–1000 IU/mL). Administration of recombinant vaccine to patients with low HBsAg levels resulted in more encouraging aging results and correlated with a further decrease in HBsAg titers and undetectable HBsAg in certain individuals. However, the study lacked a potential control group [133]. Larger randomized trials are needed to determine whether combination vaccines can help people with low baseline HBsAg levels. The initial study retrospectively investigated the impact of vaccination in preventing the multiplication of the virus once nucleos(t)ide medication was discontinued. However, the study found no advantages of vaccination in this context [134]. Administering recombinant vaccines to patients with low levels of HBsAg resulted in more favorable outcomes and led to additional declines in HBsAg levels. In some instances, the HBsAg became undetectable. However, an essential element missing from this study was the inclusion of a forward-looking control group [133].

In order to determine the potential benefits of recombinant vaccines for individuals with low initial levels of HBsAg, it is necessary to conduct more extensive randomized studies. The reason why recombinant HBsAg vaccines might have shown inefficacy is that they solely provided one HBV antigen, whereas during acute infection, the immune system produces responses that target multiple HBV antigens [62]. An innovative approach has been developed to address this issue, using a vaccine technology that involves administering both HBsAg and HBcAg. In a randomized trial in subjects with chronic HBV, treatment with recombinant HBsAg and HBcAg achieved significant results in significantly reducing HBV DNA levels after 24 weeks of treatment, surpassing the efficacy of peginterferon alone [135]. The failure to mention the decrease in HBsAg is surprising, especially given the promising implications of these results.

### 4.8. Lipopeptide Epitope-Based Vaccine

Peptide-based vaccines are now one of the most broadly studied prophylactic as well as therapeutic healthcare treatments against a wide range of diseases. In order to improve the efficiency of vaccines in activating T cells, a novel lipopeptide vaccine called CY-1889 was developed against HBV. This innovative vaccine is epitope based and specifically designed to stimulate a specific class of T cells known as CD8+ T cells. One significant CD8+ T cell core epitope (Core18-27) related to HBV was included in the CY-1889 vaccine. This core epitope has been detected in individuals who have successfully controlled acute HBV infection. When CY-1889 was administered to patients chronically infected with HBV, it was found that this immunization significantly increased the levels of CD8+ T cells specific to the HBV core epitope. This boost in the CD8+ T cell response was demonstrated in healthy volunteers. However, the potency of CD8+ T cell responses was comparatively lower, and there were no detectable changes in the amount of HBV DNA in these individuals [26]. The finding that an immunological approach that targets a specific epitope would only be effective in those carrying the HLA alleles linked to that epitope is a significant limitation. As a result, the immune response may be less effective, and the development of immunological escape variations may be helpful. Lai Wei et al. have created a liposome-based nanoparticle vaccine (εPA-44) through coordinated adjuvant, delivery mechanism, and immunogen optimization. Instead of using natural HBV antigens as an immunogen to prevent undesirable immune responses (like Th2), break tolerance, and increase therapeutic immunity, they suggested using a de novo-designed antigen (mimogen) that induces dominant and potent protective immune responses mimicking acute HBV infection. Liposomes have been used as a particle delivery system and additional adjuvant to activate cytotoxic T lymphocyte (CTLs) through cross-presentation, the benefits of which have been confirmed in subunit vaccines. Phase 1 and pilot phase 2 trials have demonstrated that 1200 μg of PA-44 was well tolerated, and that 300 μg and 600 μg, respectively, were the least efficacious dosages to produce an HBV-specific CTL response or lower the viral load. Here, in a two-stage phase 2 research, we documented the safety and effectiveness of 600 μg and 900 μg εPA-44 in patients with HBeAg-positive CHB [136].

### 4.9. Nanoparticle-Based Vaccines

Nanoparticles have proven to be an excellent platform to enhance the immunogenicity of an antigen to elicit a high-level and sustained anti-HBV immune response [137]. Recombinant protein immunizations, with unadulterated fixings and great safety, are slowly replacing a few attenuated and inactivated vaccines in clinical practice. In recent years, several research groups have invested in the development of nanoparticles/microparticles made of biodegradable and biocompatible polymers as vaccine delivery systems with the aim of inducing both humoral and cellular immune responses. Many strategies to direct subunit immunization are currently being evaluated. Currently, nanoparticles are the main technology used in the production of vaccines. These vaccines, referred to as nanovaccines, have unparalleled advantages compared to traditional vaccines [138]. Due to their size and pathogen-like appearance, nanovaccines can mimic a natural infection with pathogens and are easy to be endocytosed by antigen present cells (APCs). In order to improve antigen presentation, nanovaccines can co-deliver antigen and adjuvant to specific APCs. Most significantly, size-dependent drainage causes nanovaccines to passively drain to the lymph nodes (LNs), which is thought to be a valuable characteristic for the effectiveness of the vaccine. Flash nano complexation (FNC) technology was used to develop a scalable and size-uniform enterovirus 71 (EV71) VP1 subunit nano-vaccine using biocompatible chitosan, heparin, and molecular adjuvant unmethylated cytosine-guanine oligodeoxynucleotides (CpG), an agonist of toll-like receptor 9. The resulting nanovaccine, a potential EV71 vaccine candidate, targeted LNs and induced potent humoral and cellular immunity [139]. Nanovaccines successfully triggered innate immunity in wild-type mice, impressively enhanced adaptive immunity and exhibited beneficial LN-targeting ability [140]. Nanoparticle vaccine induces a potent anti-preS1 response that effectively clears the virus and partially converts the serological status in a chronic HBV mouse model, providing a potential vaccine approach that can be used for the functional treatment of chronic hepatitis B.

Currently, anti-HBV gene therapy strategies mainly include ribozyme, RNA interference (RNAi), and gene-editing technology. However, several parameters influence the in vivo efficacy of HBV-based antigen drugs. For example, these drugs are easily broken down by enzymes in the blood and liver and are hydrophilic macromolecules with a negative charge that cannot penetrate cell membranes effectively. Therefore, effective vectors are needed for the therapeutic use of gene-based drugs. Viral vectors exhibit high transfection efficiency; however, viral proteins can induce an immune response and there are potential safety concerns such as viral replication in the body [141]. The majority of clinical trials of viral vector gene therapy are still in clinical phase I (safety testing phase). Therefore, non-viral gene vectors have attracted great attention and have been applied in the research of many anti-HBV gene-based drugs. Nanovaccines offer incomparable advantages over traditional vaccines, such as improved antigen presentation, passive access to the immune response and its epicenter, and disease-identical size and morphology [142].

### 4.10. HBV Vaccines in Clinical Trials

Therapeutic vaccination has been proposed to further improve the efficacy of existing antiviral regimens, as robust, adaptive, and innate immune responses are required to adequately clear chronic HBV infection and that HBV can impair these responses. Several studies have been reported for the development of therapeutic vaccines against the HBV. In order to improve our understanding of HBV therapeutic vaccination, some of these research studies are summarized in Table 4. Vaccine efficacy looks at how immunization affects people in the real world, both individually and as a group. The generalizability of trial results to the vaccine’s effectiveness in the real world is thus fundamentally constrained, even if randomized clinical trials may assess a vaccine’s safety and effectiveness under ideal research settings [143]. The multidose, 6-month schedule for approved three-dose vaccines (such as Engerix-B^®^ (HepB-Alum); GlaxoSmithKline, Research Triangle Park, NC, USA; and Recombivax HB^®^; Merck Sharp & Dohme, Whitehouse Station, NJ, USA) limits the effectiveness of the vaccine against HBV infection because vaccine series completion adherence is low, with only 22–60% of series initiators in the United States completing the three-dose regimen within a year [144,145]. These low rates are particularly worrisome for patient categories who are more susceptible to HBV infection, such as those with diabetes [146]. In contrast, two-dose vaccination with smooth completion of the 1-month series—HEPLISAV-B^®^ (HepB-CpG) was shown; Dynavax Technologies, Emeryville, CA, USA—can provide faster and better seroprotection than three-dose vaccines and higher completion rates. Therefore, this two-dose vaccine can prevent HBV infection more effectively than three-dose immunization at individual and community levels [146,147].

## 5. Concluding Remarks and Future Prospects

Despite the great success of HBsAg-based immunizations in inducing the production of anti-HBs and establishing immunity against HBV, vaccines have been generally ineffective when used as therapeutic agents. Vaccination has been used as a therapeutic approach to achieve long-term suppression of HBV replication and ultimately decrease the levels of HBsAg in individuals with chronic HBV. The aim of this strategy is to disrupt T-cell tolerance to HBV proteins and enhance the body’s HBV-specific T cell responses. Regrettably, studies conducted on protein, DNA, and T-cell vaccines in patients with chronic hepatitis B have not resulted in successful outcomes. This lack of success can possibly be attributed to the excessive focus on HBsAg during the development of these vaccines [152]. The use of vaccines as a viable approach in the treatment of CHB is constantly increasing, mainly due to the development of innovative DNA vaccines, heterologous primary booster vaccines, immunization targeting different HBV proteins, and new adjuvants [153]. 

Phase 1 trials of DNA vaccines such as JNJ-64300535 and INO-1800 are currently underway. INO-1800 consists of rec-DNA vaccines containing the genetic code of the HBsAg and HBcAg consensus sequences. However, DNA vaccination has not been found to significantly improve immunity [154]. The current study is investigating INO-1800 in adults with chronic hepatitis B, either as a stand-alone therapy or in combination with IL-12. Preliminary findings indicate that INO-1800 is safe, well tolerated, and generates virus-specific T cells, including CD8+ killer T cells [155]. We look forward to the results of the trial of JNJ-64300535 in conjunction with nucleos(t)ide analogues [154,156]. Multiple HBV proteins generated by various vectors combined with adjuvants have been used in vaccines that have exhibited some effectiveness in vitro and in animal models [154]. A possible therapeutic strategy to completely prevent intrahepatic HBV replication may involve the combination of several direct-acting antiviral drugs with different mechanisms of action. Multiple strategies have the potential to enhance the effectiveness of therapeutic vaccination for chronic HBV. While HBV vaccine candidates have shown immunogenicity in healthy individuals, they have consistently failed to elicit significant HBV-specific T cell responses when administered to individuals with chronic HBV [26,62]. Therefore, possible methods for improving therapeutic vaccination all seek to increase vaccine-induced T cell responses and eliminate the malfunctioning of HBV-specific T cells brought on by long-term infection. The fight against HBV is not over, but widespread use of hepatitis B vaccine is the foundation and most important tool we have to control HBV. The HBV vaccine is a significant advancement in contemporary medicine, and it should be made available to everyone who is at risk. Adequate titers must then be shown after the immunization. One area that is overlooked is the therapeutic efficacy of a vaccine for treating an uninfected patient with no symptoms. However, vaccination benefits both therapeutically and prophylactically from the treatment of pre-existing infections and the prevention of reinfection.

## Figures and Tables

**Figure 1 vaccines-11-01862-f001:**
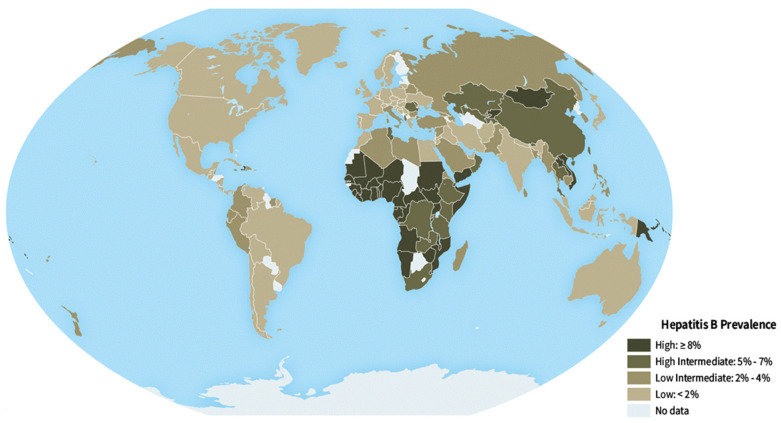
Geographical distribution of the hepatitis B disease burden worldwide across all countries and territories [8]. Variations in color intensity indicate the severity of the disease in different regions.

**Figure 2 vaccines-11-01862-f002:**
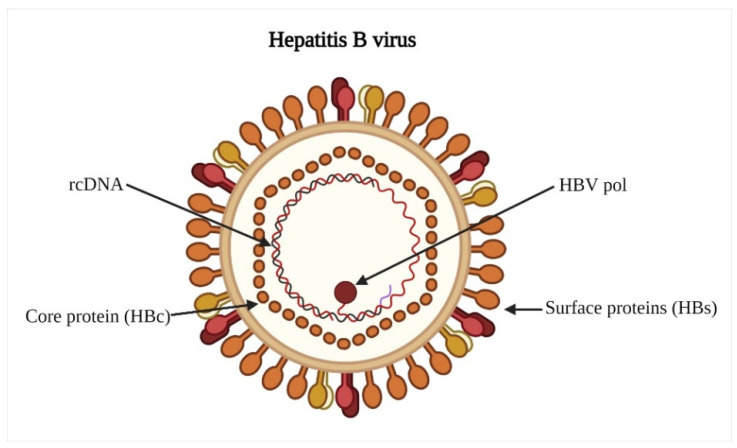
Basic structure of HBV virion. The inner protein shell of the virus, known as the main particle or “HBcAg,” contains the viral DNA and enzymes responsible for replication, commonly referred to as “DNA polymerase.” On the other hand, the outer membrane, composed of lipid and protein, is called the “surface antigen” or “HBsAg.” rcDNA stands for “relaxed circular DNA” and HBV pol stands for “HBV polymerase”.

**Figure 3 vaccines-11-01862-f003:**
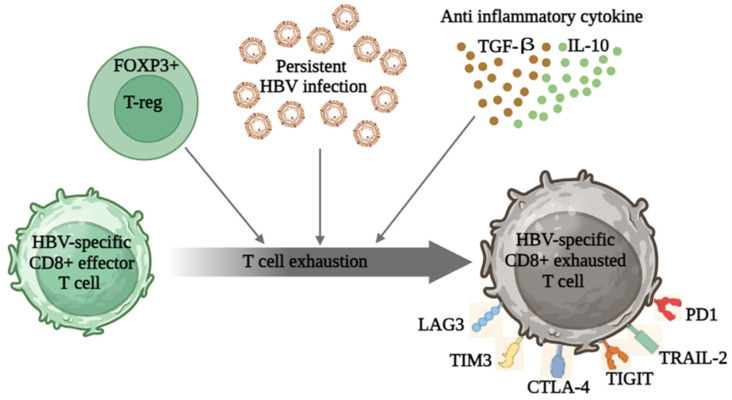
Dysregulation of immune response after HBV infection. Persistent HBV infection causes HBV-specific CD8+ T cells to exhaust themselves. The death receptor TRAIL-2 is upregulated in exhausted CD8+ T lymphocytes, which show reduced proliferation and are more likely to undergo apoptosis. When CD8+ T cells run out, their effector activity diminishes, but they remain partially activated, leading to hepatocyte injury that persists, repeated DNA damage, genomic instability, mutation accumulation, and ultimately neoplastic transformation. Furthermore, a tumor-prone immunological milieu is produced when worn-out HBV-specific CD8+ T cells express a number of inhibitory molecules, including PD-1, TIGIT, CTLA-4, TIM3, and LAG3. These factors also impair CD8+ T cells’ ability to perform immune surveillance.

**Figure 4 vaccines-11-01862-f004:**
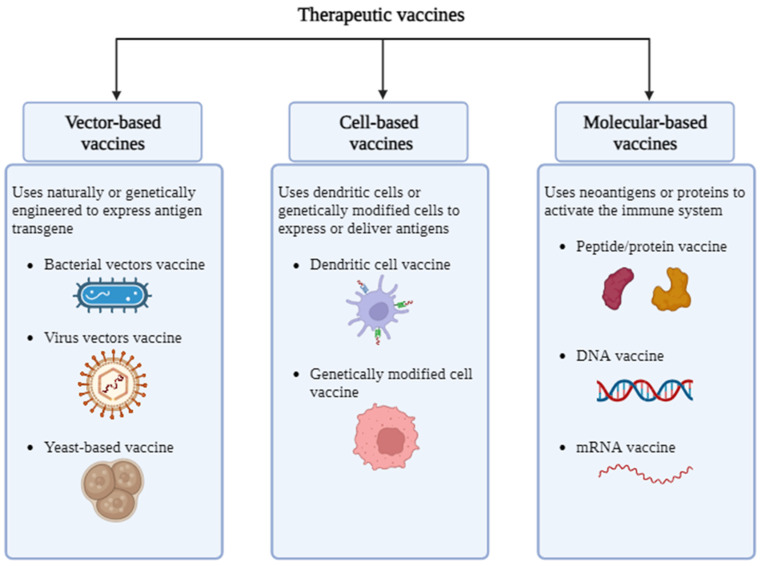
Schematic presentation of basis of therapeutic vaccines development. There are various types of therapeutic vaccines, such as vector-based vaccines, molecular-based vaccines, and cell-based vaccines.

**Figure 5 vaccines-11-01862-f005:**
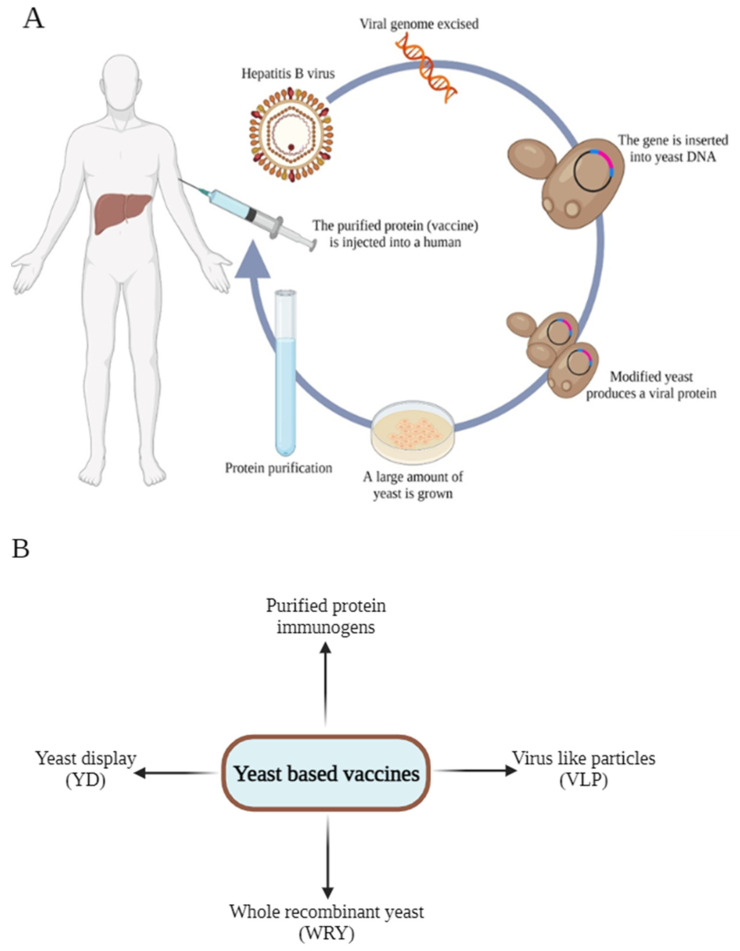
(**A**) Schematic presentation of generation and implementation of yeast HBV vaccine. (**B**) Different yeast-based platforms for vaccine development including purified protein immunogens, yeast display, whole recombinant yeast and virus like particle.

**Table 1 vaccines-11-01862-t001:** Different generations of HBV vaccine development.

Vaccine Generation	Type	Year	References
First hepatitis B Vaccine	Heat-treated form of the virus.	1971	[33]
I generation Hepatitis B Vaccine	Plasma-derived hepatitis B vaccine	1981	[34]
II generation Recombinant Vaccine	HBV DNA vaccine, expressed in yeast	1986	[34,35]
III generation Recombinant Vaccine	pre-S/S vaccines expressed in mammalian cells	1990	[34,35]
IV generation Recombinant Vaccine	Recombinant HBV vaccines with adjuvant (AS04, CpG)	2005	[36]

**Table 2 vaccines-11-01862-t002:** Certified adjuvant vaccines.

Vaccine	Adjuvant	Dose	Age	Administration	Reference
FENDRIX	3-O-desacyl-4′-monophosphoryl lipid A and aluminum phosphate.	Four doses. (There should be a gap of 1 month between the first and second, and between the second and third injections. The fourth injection is given 4 months after the third.)	15 years onwards.	Intramuscularly	[44]
ENGERIX B	Aluminum hydroxide	Three shots over a 6-month period.	Used in both pediatrics, starting with infants at birth, and adults.	Intramuscularly	[45]
HBVAXPRO	Amorphous aluminum hydroxyphosphate sulfate	At least three doses.	From birth through to 15 years of age.	Intramuscularly	[44]
HEPLISAV	Cytosine phosphoguanosine (CpG) 1018 adjuvant (HepB-CpG)	Only two doses.	Adults aged 18 years and older.	Intramuscularly	[46]

**Table 4 vaccines-11-01862-t004:** HBV vaccines in clinical trials.

Vaccine Type	Administered Regime	Adjuvant	Results Summary	Trial Phase	Trial Registration	Reference
HBsAg-HBIG	Breaking immune tolerance to HBV by modulating viral antigen processing and presentation	Alum	Serum HBV DNA decreased and normalization of liver function	III	NCT03575208 (https://classic.clinicaltrials.gov/ct2/history/NCT03575208. Accessed date: 15 February 2023)	[129]
GS-4774	Yeast-derived vaccine, includes HBsAg, HBcAg and hepatitis B X	Yeast component has been shown to have adjuvant properties and to reduce frequency and inhibitory activity of T regulatory cells	Activate an HBV-specific T cell immune response to reduce the number of cells containing HBV	II	NCT01943799 (https://clinicaltrials.gov/study/NCT01943799. Accessed date: 16 February 2023)	[115]
GS-9620	Orally active small molecule agonist of toll-like receptor 7 (TLR7)	TLR Agonists as Vaccine Adjuvants	Serum viral DNA and antigens were suppressed for an extended period of time	I	NCT01590654 (https://www.cdek.liu.edu/trial/NCT01590654. Accessed date: 16 February 2023)	[148]
Theravax (DV-601)	HBV surface antigen (HBsAg) and HBV core antigen (HcAg)	Saponin-based ISCOMATRIX adjuvant	The development of an HBV-specific interferon- γ T-cell response, an HBV-specific lymphoproliferative response, and a decrease in HBV DNA	Ib	NCT01023230 (https://clinicaltrials.gov/study/NCT01023230. Accessed date: 15 February 2023)	[149]
Nasvac	Hepatitis B surface antigen (HBsAg) and hepatitis B core antigen (HBcAg)	Without adjuvants	After a multi-TLR agonist action, activate several innate immune and signal transduction pathways	III	NCT01374308 (https://clinicaltrials.gov/study/NCT01374308. Accessed date: 16 February 2023)	[150]
INO-1800	DNA vaccine encoding HBsAg and a consensus sequence of HBcAg	Without adjuvants	Determines a virus-specific T-cell immune response	I	NCT02431312 (https://clinicaltrials.gov/study/NCT02431312. Accessed date: 16 February 2023)	
TG-1050	Adenovirus 5-based therapeutic vaccine expressing core, polymerase, and surface antigen HBV proteins	Without adjuvants	Capable of inducing HBV-specific cellular immune response and IFN- γ producing T-cells targeting 1 to 3 encoded antigens	1/1b	NCT02428400 (https://clinicaltrials.gov/study/NCT02428400. Accessed date: 19 February 2023)	[151]
HB-110	2nd-generation therapeutic adenoviral-based DNA vaccine encoding S, L, core, polymerase protein	Human IL-12 as adjuvant	Sustained CD4+ memory T cell responses were produced by long-lasting viral suppression and clear T-cell responses, particularly those that were memory-related	1/2a	NCT01641536 (https://clinicaltrials.gov/study/NCT01641536. Accessed date: 26 July 2023)	[120]
CVI-HBV-002	Activate the patient’s immune system effectively to fight and control the virus infection	Highly immunogenic L-HBsAg and powerful adjuvant L-pampo	Immune response was induced under the immune tolerance status and HBsAg particles in the blood were decreased by the antibody induced	II	NCT04289987 (https://clinicaltrials.gov/study/NCT04289987. Accessed date: 16 February 2023)	[132]

## Data Availability

The data is available in the articles included in this review.

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
