# Peer review of "HBV Vaccines: Advances and Development"

_vaccines, 2023, doi:10.3390/vaccines11121862_

Round 1

Reviewer 1 Report (New Reviewer)

Comments and Suggestions for Authors

Comments and suggestions for author of manuscript titled"HBV Vaccines: Advances and Development". The review is informative in in depth. Overall draft look nice and just need minor revision

1) Size of Figure 1 can be increased for better visibility

2) Legend of figure can be improved

3) In line 268 make H. influenza  italic

4) In line 413 write K. phaffii for Pichia pastoris as name is changed

5) Will be nice if author can include a schematic showing basic behind therapeutic vaccine

6) In fig 3 author should improve legend

7) In fig 3 , author can include one more panel showing different yeast-based platforms including VLPs, purified proteins, yeast-display, whole recombinant yeast

8) Will be nice if author can include schematic showing HBV-INDUCED IMMUNE DYSREGULATION

Comments on the Quality of English Language

Overall writing is OK

Author Response

General Comment

Comments and suggestions for author of manuscript titled"HBV Vaccines: Advances and Development". The review is informative in in depth. The overall draft looks nice and just need minor revision.

Response: We are very thankful for the constructive comments and positive feedback on our manuscript. We have revised our manuscript further based on these additional comments. We have found that these comments have substantially improved the quality of our work. We hope this revised manuscript addresses all the concerns.

  • Size of Figure 1 can be increased for better visibility.

Response: We have increased the size of Figure 1 to improve its visibility for the readers.

  • Legend of figure can be improved.

Response: We have modified the data according to reviewer’s suggestions. 

  • In line 268 make H. influenza  italic

Response: We have changed “H. influenza” to italic “H. influenza”    

  • In line 413 write K. phaffii for Pichia pastoris as name is changed

Response: Thank you for mentioning this we have modified the name according to reviewer’s suggestions.

  • Will be nice if author can include a schematic showing basic behind therapeutic vaccine

Response:  We have added a schematic diagram to show the basic behind therapeutic vaccines.

  • In fig 3 author should improve legend

Response: We have improved the legend of Figure 3.

  • In fig 3, author can include one more panel showing different yeast-based platforms including VLPs, purified proteins, yeast-display, whole recombinant yeast.

Response: Thank you for the constructive comments we have added yeast-based platforms along with Figure 3.

  • Will be nice if author can include schematic showing HBV-INDUCED IMMUNE DYSREGULATION

Response: Thank you for pointing this out. We have added a figure to show the HBV- induced immune dysregulation.

Reviewer 2 Report (Previous Reviewer 3)

Comments and Suggestions for Authors

The reviewer's suggestions have been addressed.

Author Response

Comments. The reviewer's suggestions have been addressed.

Response: We appreciate your insightful criticism and comments on our revised manuscript. We found that the standard of our work has significantly increased as a result of these remarks.

Reviewer 3 Report (New Reviewer)

Comments and Suggestions for Authors

This manuscript is a review of the evolution, present and future of HBV vaccines. It can be useful as reference for researchers and practitioners.

The work is well structured and it describes de different types of vaccines. 

I think that the section 2  explaining Hepatitis B virus, could be shorter, as it is not strictly necessary to give so many details. Potential readers of this review already know this information.

Author Response

This manuscript is a review of the evolution, present and future of HBV vaccines. It can be useful as reference for researchers and practitioners.

The work is well structured and it describes de different types of vaccines.

I think that the section 2 explaining Hepatitis B virus, could be shorter, as it is not strictly necessary to give so many details. Potential readers of this review already know this information.

Response: Thank you for taking the time to review our manuscript. We appreciate your valuable feedback and suggestions. We have shortened the "Hepatitis B virus" in section 2 to make it more engaging for readers.

Reviewer 4 Report (New Reviewer)

Comments and Suggestions for Authors

The authors extensively review the development of various HBV vaccines and the challenges. This research field is important, and the review covered broad information with a total of 174 references listed. However, the manuscript has many places inaccurate, disorganized, repeating, confusing, or even misleading in some sections. Even the abstract is poorly organized. Some of the examples are listed.

1. In Abstract: Line 21-31: “However, this success is now in jeopardy due to the breakthrough infections caused by HBV variants with mutations in the S gene , high maternal viral loads, and virus-induced immunosuppression. …  The existing vaccines have proven to be highly effective in preventing HBV infection, but ongoing research aims to improve their efficacy, duration of protection, and accessibility.”

Each of these sentences is fine, but the order of these sentences or ideas is poorly organized. The first said “this success is now in jeopardy”, then followed by “vaccines have proven to be highly effective”. This is very confusing.

In line 34-37: “The first hepatitis B vaccine approved was developed by purifying the hepatitis B surface antigen (HBsAg) from the plasma of asymptomatic HBsAg carriers. Subsequently, recombinant DNA technology led to the development of the recombinant hepatitis B vaccine.”

These are the background information of the hepatitis B vaccines, which should be in the front, not in the end of the abstract.

2. Line 89-92: “Approximately one-third of individuals have encountered this virus, resulting in a substantial loss of 1.5 million lives annually. Notably, most infected people do not show any symptoms. Alarmingly, over 780,000 deaths occur each year due to complications arising from HBV infection.”

1.5 million or 780,000 deaths? Confusing.

3. Line 133-134: “The virus in hepatocellular tissue expressed 17 proteins in a distinct manner, with 10 being up-regulated and 7 being downregulated.”

Not clear. Viral proteins or cellular proteins? How up and down regulations are termed or compared?

4. Line 309-313: “Injecting vaccines is widely recognized as a method that often fails to effectively induce mucosal immunity. Consequently, these vaccines may exhibit reduced efficacy in fighting infections occurring at mucosal surfaces. Theoretically speaking, compared to administering vaccines directly to mucosal surfaces, injecting them can induce a more potent mucosal immune response.”

The statement first is that injecting fails to effectively induce mucosal immunity, but ending up that injecting can induce a more potent mucosal immune response. What you really want to say.

5. Line 451-456: “Adenoviruses are highly sought-after for the development of gene therapies targeting HBV due to their strong affinity for the liver. …  AdVs are desirable for the development of HBV-specific gene treatments due to their strong liver tropism.” Repeating.

6. Line 355-357: “Therapeutic vaccines for chronic HBV infection have demonstrated the tremendous difficulties in reestablishing a strong HBV-specific immunity. While therapeutic vaccination has been shown to partially induce T-cell and B-cell tolerance, long-term clinical benefits or HBV infection control have seldom been seen [67].” Check these sentences. Confusing.

7. Line 394-396: “To increase T cell response, even the more immunogenic therapeutic vaccines now under development will probably need to be used in conjunction with specialized immunotherapies.”

What is more immunogenic (efficient?)

8. “Figure 3. Hepatitis B vaccine development.”

But it is about using yeast to develop vaccine. Inaccurate.

9. Line 511-512: “successfully triggered an immune response against anti-HBs in mice, monkeys, and chimpanzees.” (response against HBs or anti-HBs)

10. Line 633-634: “One possible explanation for the failure of these trials is the assumption that vaccination methods needed to successfully prevent HBV infection would be significantly different from those required to cure established chronic HBV infection.”

Do you want to say: … vaccination methods needed to cure established chronic HBV infection would be significantly different from those required to successfully prevent HBV infection. (?)

11. Line 643-655: “Two small trials have assessed the effectiveness of the recombinant HBsAg vaccination in individuals who have a disease that lacks the specific antigen, suppressed HBV DNA, and low levels of antigens (between 100-1000 IU/ml). Recombinant vaccine administration to patients with low HBsAg levels led to more encouraging outcomes and was correlated with further decrease in HBsAg titers and undetectable HBsAg in certain individuals. The study, however, lacked a prospective control group [145]. To determine whether recombinant vaccines can help those with low baseline HBsAg levels, larger randomized studies are needed. The initial study retrospectively investigated the impact of vaccination in preventing the multiplication of the virus once nucleos(t)ide medication was discontinued. However, the study found no advantages of vaccination in this context [146]. Administering recombinant vaccines to patients with low levels of HBsAg resulted in more favorable outcomes and led to additional declines in HBsAg levels. In some instances, the HBsAg became undetectable. However, an essential element missing from this study was the inclusion of a forward-looking control group [147].”

Repeated discussing the same study, with the same reference but giving different numbers (reference 145 and 147 is in fact the same one).

Others minor examples:

Line 22: “in the S gene , high maternal viral loads”

Line 322: “DNA carrying pRc/ CMV-HBs”

Line 416: “illnesses [80,81].One advantage”

Line 474: “against the pathogen. [93,94]. It is worth”

Again, these are some examples, many more may be found in the entire paper and need to be worked on. The entire manuscript should be carefully edited, some parts may need to be rewritten. However, the information covered in this manuscript would be helpful to investigators in the field. The information is just need to be better organized and presented.

Comments on the Quality of English Language

need some work.

Author Response

The authors extensively review the development of various HBV vaccines and the challenges. This research field is important, and the review covered broad information with a total of 174 references listed. However, the manuscript has many places inaccurate, disorganized, repeating, confusing, or even misleading in some sections. Even the abstract is poorly organized. Some of the examples are listed.

Response: We appreciate for taking the time to review our manuscript. We value your insightful comments and recommendations, and we have revised the manuscript accordingly in light of your comments. We have revised our manuscript accordingly and feel that your comments helped clarify and improve our manuscript.

  1. In Abstract: Line 21-31: “However, this success is now in jeopardy due to the breakthrough infections caused by HBV variants with mutations in the S gene, high maternal viral loads, and virus-induced immunosuppression. …  The existing vaccines have proven to be highly effective in preventing HBV infection, but ongoing research aims to improve their efficacy, duration of protection, and accessibility.”

Each of these sentences is fine, but the order of these sentences or ideas is poorly organized. The first said “this success is now in jeopardy”, then followed by “vaccines have proven to be highly effective”. This is very confusing.

Response: Thank you for mentioning this we have rearranged the abstract to make to more precise and easier to read.

In line 34-37: “The first hepatitis B vaccine approved was developed by purifying the hepatitis B surface antigen (HBsAg) from the plasma of asymptomatic HBsAg carriers. Subsequently, recombinant DNA technology led to the development of the recombinant hepatitis B vaccine.

These are the background information of the hepatitis B vaccines, which should be in the front, not in the end of the abstract.

Response: We have rearranged the abstract.

  1. Line 89-92: “Approximately one-third of individuals have encountered this virus, resulting in a substantial loss of 1.5 million lives annually. Notably, most infected people do not show any symptoms. Alarmingly, over 780,000 deaths occur each year due to complications arising from HBV infection.”

1.5 million or 780,000 deaths? Confusing.

Response: Thank you for pointing this out we have changed the data.

  1. Line 133-134: “The virus in hepatocellular tissue expressed 17 proteins in a distinct manner, with 10 being up-regulated and 7 being downregulated.”

Not clear. Viral proteins or cellular proteins? How up and down regulations are termed or compared?

Response: Thank you for pointing this, here we have mentioned Hepatitis B virus proteins and we have revised the data in the manuscript.

  1. Line 309-313: “Injecting vaccines is widely recognized as a method that often fails to effectively induce mucosal immunity. Consequently, these vaccines may exhibit reduced efficacy in fighting infections occurring at mucosal surfaces. Theoretically speaking, compared to administering vaccines directly to mucosal surfaces, injecting them can induce a more potent mucosal immune response.”

The statement first is that injecting fails to effectively induce mucosal immunity, but ending up that injecting can induce a more potent mucosal immune response. What you really want to say.

Response: We have revised the statement to make it clear to understand.

  1. Line 451-456: “Adenoviruses are highly sought-after for the development of gene therapies targeting HBV due to their strong affinity for the liver. …  AdVs are desirable for the development of HBV-specific gene treatments due to their strong liver tropism.” Repeating.

Response: We have removed the repeated sentence.

  1. Line 355-357: “Therapeutic vaccines for chronic HBV infection have demonstrated the tremendous difficulties in reestablishing a strong HBV-specific immunity. While therapeutic vaccination has been shown to partially induce T-cell and B-cell tolerance, long-term clinical benefits or HBV infection control have seldom been seen [67].” Check these sentences. Confusing.

Response: Thank you for mentioning this we have modified the data.

  1. Line 394-396: “To increase T cell response, even the more immunogenic therapeutic vaccines now under development will probably need to be used in conjunction with specialized immunotherapies.”

What is more immunogenic (efficient?)

Response: Thank you for mentioning this. Immunogenic means “causing or producing immunity or an immune response”. The development of therapeutic vaccines is in process that can produce strong immune response.

  1. “Figure 3. Hepatitis B vaccine development.”

But it is about using yeast to develop vaccine. Inaccurate.

Response: Thank you for pointing this out that we have modified the legend of Figure 3.

  1. Line 511-512: “successfully triggered an immune response against anti-HBs in mice, monkeys, and chimpanzees.” (response against HBs or anti-HBs)

Response: Thank you pointing this out, it is “anti-HBs”.

  1. Line 633-634: “One possible explanation for the failure of these trials is the assumption that vaccination methods needed to successfully prevent HBV infection would be significantly different from those required to cure established chronic HBV infection.”

Do you want to say: … vaccination methods needed to cure established chronic HBV infection would be significantly different from those required to successfully prevent HBV infection. (?)

Response: we have modified the data according to the reviewer’s suggestions.

  1. Line 643-655: “Two small trials have assessed the effectiveness of the recombinant HBsAg vaccination in individuals who have a disease that lacks the specific antigen, suppressed HBV DNA, and low levels of antigens (between 100-1000 IU/ml). Recombinant vaccine administration to patients with low HBsAg levels led to more encouraging outcomes and was correlated with further decrease in HBsAg titers and undetectable HBsAg in certain individuals. The study, however, lacked a prospective control group [145]. To determine whether recombinant vaccines can help those with low baseline HBsAg levels, larger randomized studies are needed. The initial study retrospectively investigated the impact of vaccination in preventing the multiplication of the virus once nucleos(t)ide medication was discontinued. However, the study found no advantages of vaccination in this context [146]. Administering recombinant vaccines to patients with low levels of HBsAg resulted in more favorable outcomes and led to additional declines in HBsAg levels. In some instances, the HBsAg became undetectable. However, an essential element missing from this study was the inclusion of a forward-looking control group [147].”

Repeated discussing the same study, with the same reference but giving different numbers (reference 145 and 147 is in fact the same one).

Response: We are sorry for this mistake, we have revised the references.

Others minor examples:

Line 22: “in the S gene , high maternal viral loads”

Response: We have modified the data.

Line 322: “DNA carrying pRc/ CMV-HBs”

Response: We have modified the data.

Line 416: “illnesses [80,81].One advantage”

Response: We have modified the data.

Line 474: “against the pathogen. [93,94]. It is worth”

Response: We have modified the data.

Again, these are some examples, many more may be found in the entire paper and need to be worked on. The entire manuscript should be carefully edited, some parts may need to be rewritten. However, the information covered in this manuscript would be helpful to investigators in the field. The information is just need to be better organized and presented.

Response: Very thanks for your suggestion. We have done the language editing by the help of native speaker “Yofre C, University of Portsmouth”. We have also attached the language editing certificate.

Round 2

Reviewer 4 Report (New Reviewer)

Comments and Suggestions for Authors

no further comments

This manuscript is a resubmission of an earlier submission. The following is a list of the peer review reports and author responses from that submission.

Round 1

Reviewer 1 Report

Comments and Suggestions for Authors

It is an articulated work with a claim to completeness, which it does not reach.

The albeit justified structure in chapters that deal with the numerous and different types of vaccination strategies, reasonably edited by different authors with specific skills, suffers from repetitions of concepts already expressed and does not result well amalgamated.

There is a complete lack of detailed efficacy data on existing vaccines, in use and/or in an advanced state of preparation, with references to bibliography that is not always easily obtainable.

A complex and detailed review of this type must provide comprehensive data.

Moreover, and very important issues, 

an exhaustive analysis of the problem of escape mutants, which are just mentioned, is missing.

the strategies that can be used for low or non-responder patients must be exhaustively analysed.

the role of the different viral genotypes is not addressed at all

the role of host genetics and racial differences is not considered at all.

Throughout the work, data are lacking to interpret and classify the different responses and efficacy, according to the current and available laboratory analyses

In detail:

Line 23: rephrase

Line 40: rephrase

Line 68: provide literature

Line 77: provide more data about different countries

Line 218: edit

Line 236: provide literature

Line 264: rephrase

Line 266: edit

Line 269: rephrase

Lines 280-290: concepts already expressed, to be moved above

Line 338: add a comment about blood transmission and IgG1-2, Th1-2 response

Line 361: more details about this model

Line 363: move to a more appropriate section

Line 370: unclear

Line 384-403: too long and unnecessary

Line 450: add more details

Line 523: there are no similar evaluations for the vaccines described above

Line 528: Why do you only talk about it here and now?

Line 623 provide literature

Line 663: add more details

Line 696: rephrase

Line 719: rephrase

Line 761-796: redundant

Author Response

We are grateful to the reviewer for their insightful comments on our paper. We have been able to incorporate changes to reflect most of the suggestions provided by the reviewers. We have highlighted the changes within the manuscript. Here is a point-by-point response to the reviewers' comments and concerns. Detailed response to reviewer’s comments is in attached word file.  Please see the attached file for detailed responses.

Thank you

Reviewer 2 Report

Comments and Suggestions for Authors

The Review titled “HBV Vaccines: Advances and Development” by Faisal Mahmood et al., is a comprehensive overview on HBV vaccine development. The authors describe HBV vaccine development in detail, including new technologies and cutting-edge applications for HBV functional therapy. Furthermore, HBV cycle and pathophysiology are described in the review. The present review is extremely accurate and comprehensive. These aspects may have influenced the length of the manuscript, which could be considered too long. Nevertheless, the divisions in chapters with informative headings makes the reading easier.

Some minor revisions are needed:

Abstract

Line 23: “in the market including,” the sentence is truncated.

Main Text

Lines 41-42: please rephrase to make the sentence more comprehensible. 

Line 44: be consistent when referring to anti-HBs. Sometimes in the text they are indicated as anti-HB

Line 56: the acronym HBV is already used above in the text without any explanation of the acronym. 

Line 61: please clarify the term “vertical sexual activity”.

Line 74-76: please rephrase the sentence to make it clearer.

Lines 79-100: when the authors describe HBV viral cycle, the possibility of HBV-DNA integration in the host genome should be cited, including also appropriate reference(s).

Line 131: cccDNA acronym already explained above in the text.

Lines 157-159: this seems a repetition of the sentence from 154 to 157.

Line 207: do the authors mean “yeast-specific antigens”? Not antibodies?

Line 265-266: there is a repetition in the sentence.

Line 272: Do the authors mean “a higher rate of seroprotection”? Please rephrase.

Line 276-277: The paragraph starts with an unnecessary introduction on HBV related diseases. 

Line 277: please verify the term “populace”.

Line 280-282: Unnecessary recall to HBV modes of transmission.

Line 288-289: Unnecessary recall to HBV clinical aspects.

Line 317: “A recent…”, the subject of the sentence is missing.

Line 336: Is alum an abbreviation for aluminum? 

Line 778-780: This sentence seems a repetition of the sentence in lines 772-774.

Comments on the Quality of English Language

Minor editing of English language is needed.

Author Response

(The authors gave the same response as above.)

Reviewer 3 Report

Comments and Suggestions for Authors

The authors have reviewed the existing literature on Hepatitis B vaccine, However the manuscript is not very focussed as it discusses the entire  Hepatitis B.

Suggestions,

The manuscript needs to be condensed and more focussed.

Clinical trial outcomes of recent Hep B vaccines need to be discussed .

Diagrams need to be cited appropriately to avoid copyright issues.

The number of references may be reduced  with only recent, pertinent citations. 

Author Response

(The authors gave the same response as above.)

Reviewer 4 Report

Comments and Suggestions for Authors

The review article by Mahmood et al., entitled “HBV Vaccines: Advances and Development” highlights the recent development in the field of HBV vaccine. This is a very poorly written article, which make it difficult to read and understand. Although authors have tried to highlight different aspects of vaccine development against HBV, there are many sections of the article which are unnecessarily lengthy for example in the section Adenoviral vaccines, there is no need of details about adenovirus. Following are some of my comments about the article.

Comments to Authors

1) In the abstract, page -1 line 19-20, what is the meaning of HB? The prevalence of acute and chronic HB, liver cirrhosis, and HCC have significantly decreased as a result of the introduction of universal HBV vaccination programs.

2) In the abstract, page-1 line 23, the mentioned phrase is not correct. “Various types of licensed vaccines are available in the market including, but still there is a need of more research to develop more and more effective HBV vaccine”.

3) In the introduction, page-1 line 41 and 42, the mentioned sentence is not clear. “HBV immunization is essential as it can effectively prevent HBV infection by ensuring the potential host possesses ample surface antibodies against HBV (anti-HBs)”.

4) Figure-1 titled Worldwide prevalence patter of HBV, authors need to explain the figures in figure legend and figure is hardly visible. Authors also should label the region in the map.

5) In the introduction, page-2, line 61, authors need to clarify what is the meaning of vertical sexual activity. “Humans are the only known naturally occurring host [10], additionally, infection may be spread through the use of common syringes, vertical sexual activity, and infection through various body fluids [11]”.

6) In the figure 2, legend, page 3, line -105, what is the meaning of this sentence? Additionally, relaxed circular DNA is known as HBV pol or HBV polymerase. Relaxed circular DNA is not HBV polymerase.

7) Figure 2 represent the HBV virion structure but in the text, authors are trying to explain the life cycle of HBV, they should either change or add a figure of HBV life cycle.

8) In the section Hepatitis B Virus, page-3, line 111, what is the meaning of mentioned line. “Most people who contract HBV throughout adulthood experience a strong immune response that clears the virus after an acute illness”.

9)  In the section Hepatitis B Virus, page-3, line 115, what is the meaning of discharged variant of the core protein? “The clinical stages of chronic HBV encompass the assessment of hepatitis B e antigen (HBeAg), which is a discharged variant of the core protein, along with evaluating HBV DNA levels and liver transaminase values”.

10) Page -6, line- 263 -264 sentence is wrong. “One of them, HBsAg-enhanced immunogenicity vaccine adjuvanted with aluminum phosphate and 3-O-deacyl-4'-monophosphoryl lipid A (MPL), has been developed and used to prevent HBsAg in many countries.

11) Similarly, many mistakes are present throughout the article in terms of writing, proper use of English grammar and words, which makes hard to read and understand the content.

Comments on the Quality of English Language

Major English editing is required in order to read and understand the article.

Author Response

(The authors gave the same response as above.)

Reviewer 5 Report

Comments and Suggestions for Authors

The article entitled “HBV Vaccines: Advances and Development” has tried to provide a comprehensive view of vaccines against the hepatitis B virus (HBV). This subject has been widely discussed in hundreds of literatures and hundreds of publications are available only in PubMed. The article lacks lovely, poorly organized, and the points of discussion remain elusive and of almost no clinical significance. The specific points are mentioned below:

1.      The article is outdated as the fundamental reference of HBV prevalence has been cited by showing an article from 2012 (Reference 1). An updated article should be based on updated references.

2.      The Abstract is extremely confusing. The authors described the prophylactic vaccine. But, they mentioned that “The goal of this type of immunization is to trigger an immunological response in the host, which will stop HBV replication”. The prophylactic vaccine is not intended to stop HBV replication, rather it induces antibody that protects from future infection. Thus, the compilation of the article is defective.

3.      The aim and objectives of the study have not been properly cited. As there are several articles about this subject, the authors should clarify what is the target of the present article. In fact, they should mention what has been explained in allied articles and what should be explained more about the topic. How important are these queries? And, how those have been addressed?

4.      Without resolving these essential elements, the article is a repetition of the previous publications with limited new contributions.

5.      At present, vaccines against HBV are not preventive in nature. Rather, several therapeutic vaccines have been developed now to tackle chronic HBV infection as therapeutics. An article of this nature should enroll both preventive and therapeutic vaccines properly.  

Author Response

(The authors gave the same response as above.)

Round 2

Reviewer 3 Report

Comments and Suggestions for Authors

The manuscript is still very long and the references are too many . It may be reduced to the number recommended by the journal

Author Response

General Comment

The manuscript is still very long, and the references are too many. It may be reduced to the number recommended by the journal.

Response: Very thanks to the reviewer’s above constructive suggestions. We have reduced the length of the manuscript to make it more precise and also reduced the number of references.

Reviewer 4 Report

Comments and Suggestions for Authors

I have no further comments.

Comments on the Quality of English Language

 Moderate editing of English language required.

Author Response

Moderate editing of English language required.

Response: Very thanks for your suggestion. We have done the language editing by the help of native speaker “Yofre C, University of Portsmouth”. We have also attached the language editing certificate.

Reviewer 5 Report

Comments and Suggestions for Authors

The revised version of the article has been carefully assessed. The authors have responded to the queries of the Reviewer. However, the responses are neither satisfactory nor objective-oriented. A review article of this nature, the title of which is” HBV Vaccines: Advances and Development” should be composed of all sorts of developments regarding prophylactic vaccines and therapeutic vaccines for hepatitis B virus. The fundamental design of the article has not been developed in that manner. Adding some sentences in some areas does not provide a viable manuscript of this nature.

Author Response

General Comment

The revised version of the article has been carefully assessed. The authors have responded to the queries of the Reviewer. However, the responses are neither satisfactory nor objective-oriented. A review article of this nature, the title of which is “HBV Vaccines: Advances and Development” should be composed of all sorts of developments regarding prophylactic vaccines and therapeutic vaccines for hepatitis B virus. The fundamental design of the article has not been developed in that manner. Adding some sentences in some areas does not provide a viable manuscript of this nature.

Response: Thanks for the reviewer’s comments. Our review article is about the advances and development of HBV vaccines, we have tried our best to summarize the developments of the HBV vaccine. We have added data about therapeutic vaccines in a separate heading also and tried to explain the development of HBV therapeutic vaccines.  All the authors have revised the manuscript again and approved the manuscript in its present form, we can work furthermore about the therapeutic and prophylactic vaccines for a separate manuscript as this topic is very diverse and broad. So, it needs more attention to write a focused manuscript.